# Safety Profile of Medicines Used for the Treatment of Drug-Resistant Tuberculosis: A Descriptive Study Based on the WHO Database (VigiBase^®^)

**DOI:** 10.3390/antibiotics12050811

**Published:** 2023-04-25

**Authors:** Alemayehu Lelisa Duga, Francesco Salvo, Alexander Kay, Albert Figueras

**Affiliations:** 1Doctoral School Societies, Politics, Public Health, Pharmacoepidemiology and Pharmacovigilance, University of Bordeaux, 33300 Bordeaux, France; 2National Pharmacovigilance Center, Eswatini Ministry of Health, Mbabane H100, Eswatini; 3Baylor College of Medicine Children’s Foundation-Eswatini, Mbabane H100, Eswatini; 4Baylor College of Medicine, Houston, TX 77030, USA

**Keywords:** drug-resistant tuberculosis, pharmacovigilance, VigiBase^®^, adverse drug reaction

## Abstract

Background: The introduction of new drugs that increase the usage of repurposed medicines for managing drug-resistant tuberculosis (DR-TB) comes with challenges of understanding, properly managing, and predicting adverse drug reactions (ADRs). In addition to the health consequences of ADRs for the individual, ADRs can reduce treatment adherence, thus contributing to resistance. This study aimed to describe the magnitude and characteristics of DR-TB-related ADRs through an analysis of ADRs reported to the WHO database (VigiBase) in the period from January 2018 to December 2020. Methods: A descriptive analysis was performed on selected reports from VigiBase on the basis of medicine-potential ADR pairs. The ADRs were stratified by sex, age group, reporting country, seriousness, outcome of the reaction, and dechallenge and rechallenge. Results: In total, 25 medicines reported to be suspected individual medicines or as a fixed-dose combination in the study period were included the study. Pyrazinamide (*n* = 836; 11.2%) was the most commonly reported medicine associated with ADRs, followed by ethionamide (*n* = 783; 10.5%) and cycloserine (*n* = 696; 9.3%). From the report included in this analysis, 2334 (31.2%) required complete withdrawal of the suspected medicine(s), followed by reduction of the dose (77; 1.0%) and an increased dose (4; 0.1%). Almost half of the reports were serious ADRs mainly caused by bedaquiline, delamanid, clofazimine, linezolid, and cycloserine that are the backbone of the DR-TB treatment currently in use. Conclusions: A third of the reports required medication withdrawal, which impacts treatment adherence and ultimately leads to drug resistance. Additionally, more than 40% of the reports indicated that ADRs appeared two months after the commencement of treatment, thus it’s important to remain alert for the potential ADRs for the entire duration of the treatment.

## 1. Introduction

Tuberculosis (TB) remains the world’s leading cause of death from infectious agents, exceeding human immunodeficiency virus/acquired immune deficiency syndrome (HIV/AIDS) [1,2]. Globally, the incidence of drug-resistant TB (DR-TB) has increased since the introduction of chemotherapy for the treatment of *Mycobacterium tuberculosis* in 1943 [3]. In 2018, 10 million people developed TB, and 1.5 million died from the disease at a worldwide level [1]. Moreover, about 500,000 new cases of multidrug- and rifampicin-resistant tuberculosis (MDR/RR-TB) are estimated to emerge annually; only one in three cases were reported by all countries in 2018 [1]. Despite significant progress in diagnostics, more effective medicines for earlier detection, and higher success rates among patients with MDR/RR-TB in a number of countries, the overall treatment success rate reached only 56% in 2018 [4].

TB that is resistant to at least rifampicin (R) and isoniazid (INH) has posed a threat to all efforts to control TB [5]. DR-TB can occur when the drugs used to treat TB are misused or mismanaged [6], and it can be transmitted directly. Common causes of multidrug-resistant TB can also be related to the health care system, inadequate/poor treatment regimens, poor adherence, lack of information on treatment, primary transmission, and the side effects of treatment [7,8].

The management of multidrug-resistant TB requires multiple medicines for a longer duration of treatment with drugs that are more expensive and difficult to tolerate [9]. In recent years, two newly introduced (bedaquiline and delamanid) and two repurposed (clofazimine and linezolid) medicines have been introduced to treat DR-TB [10,11]. According to the WHO’s consolidated guidelines on DR-TB, treatment regimens are organized into three groups for the purposes of constructing a regimen while balancing efficacy and safety [12,13]: Group A (levofloxacin, moxifloxacin, bedaquiline, and linezolid); Group B (clofazimine, cycloserine, and terizidone); and Group C (ethambutol, delamanid, pyrazinamide, imipenem-cilastatin, meropenem, amikacin, streptomycin, ethionamide, prothionamide, and p-aminosalicyclic acid (PAS)) [12].

The principle of designing a regimen for treating DR-TB is to individualize it, guided by the results of a drug-susceptibility test (DST) prior to the treatment’s initiation, the TB treatment history, a contact DST, and co-morbidities [13]. Regimens are designed with at least four to five effective TB medicines. Group A medicines are prioritized, followed by Group B; Group C medicines are added to complete the regimens when agents from Groups A and B cannot be used [14,15].

Pharmacological interventions sometimes carry inherent risks, which include ad-verse drug reactions (ADRs) and drug interactions (DIs). The incidence of ADRs due to DR-TB drugs ranges from 20% to 90% [16,17,18]. ADRs associated with DR-TB drugs include joint pain, nausea, hearing disturbances, gastrointestinal disturbances, depression, itching, hypothyroidism, dizziness, seizures, and hepatitis [17,19,20,21,22]. About one patient in five is withdrawn from DR-TB drugs because of ADRs [23].

Collectively, ADRs represent a clinically significant problem and burden with high incidence and prevalence. The safety profile of the medicines used in patients with DR-TB is not comprehensive, as these are complicated patients and often are treated in places where there are no pharmacovigilance systems or only ones that are poorly developed. We aimed to study the profile of ADRs of the medicines used in DR-TB patients reported to the global WHO pharmacovigilance database.

## 2. Results

In VigiBase, 349,831 cases of ADRs potentially related to DR-TB drugs were identified using the predefined search criteria; of these, 342,357 ADRs were excluded because they were associated with concomitant (not suspected) and unspecified medicines, indications other than DR-TB cases, an unknown drug start date, and unknown ADR start date and ADRs not described in the report.

This study was therefore based on 7474 reports associated with 25 different DR-TB drugs that were reported as suspected individual medicines or as a fixed-dose combination. Most reports were from India (*n* = 5048; 67.5%), South Africa (446; 5.9 %), and Eswatini (304; 4.1%).

The mean and median ages of the patients were 35.6 years (SD = 15.6) and 33 years, respectively, while the most-represented groups were male patients (4030; 53.9%) and those aged 19–64 years (6435; 86.1%).

Among the 25 medicines included in these reports, pyrazinamide (*n* = 836; 11.2%) was the most commonly reported, followed by ethionamide (783; 10.5%) and cycloserine (696; 9.3%) (see Table 1).

ADRs were firstly categorized according to the classifications of the System Organ Class MedDRA terms. Gastrointestinal disorders (*n* = 1650; 22.1%) and nervous system disorders (709; 9.5) were the most frequently reported ADRs grouped by system/organ (see Figure 1). It should be noted that 739 (9.9%) reports were described as “investigations”.

With regard to specific ADRs, vomiting (*n* = 834; 11.6%) was the most frequently reported ADR, followed by arthralgia (331; 4.4%), nausea (275; 3.7%), peripheral neuropathy (216; 2.9%), and prolongation of electrocardiogram QT (204; 2.7%).

### 2.1. Seriousness of ADRs

Almost half of the reports (*n* = 3365; 45.1%) described serious ADRs. Seriousness was not different between men (4030; 53.9%) and women (3444; 46.1%). However, as indicated on the Table 2, serious ADRs were more frequent in children and adolescents (*n* = 363; 55.7%) than in adults (2822; 43.8%) and the elderly (180; 46.5%) (chi-square = 34.9; *p* < 0.001).

The majority of serious ADRs according to the terms of the MedDRA System Organ Class (SOC) were reported as gastrointestinal disorders (*n* = 406; 12.1%), followed by respiratory disorders (296; 8.8%), nervous system disorders (231; 6.9%), and ear and labyrinth disorders (212; 6.3%). Upon analyzing the seriousness of ADRs according to the preferred terms (PTs) of MedDRA, vomiting (*n* = 34; 11.2%), arthralgia (331; 4.4%), and nausea (275; 3.7%) were the top three reported serious ADRs. In terms of medicines, the majority of serious ADRs (*n* = 1846; 54.9%) were caused by bedaquiline (457; 67.4%), followed by delamanid (391; 11.6%), clofazimine (370; 11.0%), linezolid (342; 10.2%), and cycloserine (286; 8.5%) (see Figure 2). Ethionamide (577; 7.72%), pyrazinamide (568; 7.60%), cycloserine (410; 5.49%), levofloxacin (338; 4.52%), and ethambutol (335; 4.48%) caused non-serious reactions.

A cross-tabulation of the top five medicines against the top five reported serious ADRs based on the SOC is shown in Table 3.

Special mention must be made of the 554 (16.5%) reports describing investigations, which are diagnostic tests and results. As expected, most of the reports including this term were serious. Bedaquiline contributed to the majority of the serious ADRs categorized under investigation-related SOC MedDRA terms. Prolonged QT, decreased hemoglobin, increased liver function test, increased aspartate aminotransferase, increased white blood cell count, and decreased blood sodium accounted for more than 50% of the serious ADRs (based on PT) under the investigation-related medDRA SOC terms.

The most frequent individual serious ADRs according to the MedDRA PT terms are described in Table 4.

### 2.2. Duration of the Reaction and Time to the Onset of the Reaction

With regard to the timing of the ADRs’ occurrence, most ADRs (*n* = 4438; 59.3%) appeared in less than three months (a median time of 61 days, IQR: 14–161) (Figure 3). The most frequent ADRs, such as vomiting (*n* = 834; 11.2%), arthralgia (331; 4.4%), and nausea (275; 3.7%), had a mean onset of the reaction of 71, 124, and 92 days, respectively. However, 3036 ADRs (40.7%) started between 3 and 24 months after the treatment onset; the most frequent late ADRs were heavy menstrual bleeding (*n* = 1; 0.01%), premature delivery (2; 0.02%), and chronic cholecystitis (2; 0.02%), with a mean onset of the reaction of 660, 658, and 647 days, respectively.

The visual display of the time to the onset of the reaction for most of ADRs had similar patterns, with the exception of optic neuritis (*n* = 28); in this case, the reports slightly increased with longer duration of treatment (see Figure 4). In this study, the time to the onset of optic neuritis had a median time of 302 days (IQR: 223–361). The majority of the suspected medicines associated with optic neuritis were linezolid (*n* = 16; 35.5%) and ethambutol (8; 17.7%). The mean onset of optic neuritis associated with linezolid and ethambutol was 251 and 263 days, respectively.

The remaining ADRs showed a descending reporting pattern after an initial peak during the first two months after starting the treatment. Notwithstanding this, it should be noted that for arthralgia (*n* = 331 reports), after the initial peak, there were successive peaks around 5, 7, and 9 months after starting the treatment. The most frequently suspected medicines in these cases of arthralgia were pyrazinamide (*n* = 130; 39.3%) and levofloxacin (*n* = 68; 20.5%). Reports describing prolonged QT (*n* = 204) had an initial peak (*n* = 59 reports; 28.9%), followed by a plateau between the second and fourth months (*n* = 68; 33.3%) that greatly diminished after the fourth month (*n* = 58; 28.4%). Reports of deafness (*n* = 146 reports) peaked at the second month (*n* = 32; 21.9%) and maintained a slightly descending plateau until the fifth month (cumulative *n* = 59; 40.4%) and then almost disappeared after that moment.

In terms of duration, most ADRs resolved in less than two weeks after their onset (median: 7 days; IQR: 3–14). The most frequent ADRs, such as vomiting (*n* = 360; 19.2%), nausea (110; 5.8%), and pruritus (77; 4.1%), had a mean duration of less than two weeks. In contrast to this, the rarer reports describing a decrease in blood albumin, a decrease in blood calcium, and a decrease in weight (1 report each; 0.1%) took around 300 days to be resolved.

With regard to medicines, streptomycin (*n* = 1; 0.1%), PAS (7; 1.6%), and moxifloxacin (13; 5.8%) were associated with reactions that had a mean duration of less than 7 days. Terizidone (*n* = 8; 0.4%), followed by the fixed-dose combination of ethambutol + isoniazid + pyrazinamide + rifampicin (1; 0.1%) and delamanid (137; 7.3%) were associated with reactions that had the longest mean durations of 34, 37, and 61 days, respectively.

### 2.3. Outcomes of the Reactions

The current study shows that almost one-quarter (*n* = 1740; 23.3%) of the ADRs were fatal or not resolved when they were reported, while 82 (1.1%) recovered/resolved with sequelae, and 1901 (25.4%) of the ADRs were still recovering/resolving. A further analysis of the reactions’ outcomes in relation to the medicines showed that bedaquiline, clofazimine, linezolid, delamanid, and pyrazinamide were responsible for the majority of fatal outcomes of the ADRs.

We further analyzed the outcomes of serious reactions; accordingly, 435 (2.9%) of the serious ADRs were not recovered/not resolved when they were reported, while 404 (12.01%) were reported as fatal. The cross-tabulation of the ADRs’ outcomes against the seriousness of the reaction in the following Table 5 shows that more fatal outcomes of the reaction were reported for serious reactions (X^2^ = 1196.2, DF = 10, *p* < 0.001).

The association between the reactions’ outcome and sex was also analyzed. Fatal outcomes of the reaction (*n* = 263; 3.5%) and reactions that had not recovered when they were reported (730; 9.8%) were observed more often in male than in female patients.

As indicated in Figure 5 below, fatal reactions were reported more frequently among adults (*n* = 383; 6.0%) than in children and adolescents (16; 2.5 %) and the elderly (9; 2.3%). Overall, elderly patients had the poorest outcomes relative to adolescents (0–18 years) and adults (19–64 years). With regards to the medicines, more than 50% of the fatal reactions reported were associated with bedaquiline (*n* = 77; 19.1%), clofazimine (63; 15.6%), linezolid (42; 10.4%), and delamanid (39; 9.7%) (X^2^ = 822.684, DF = 110, *p* < 0.0001).

### 2.4. Discontinuation of Treatment Due to ADRs

Of the 7474 reports, 2334 (31.2%) required complete withdrawal of the suspected medicine(s), followed by reduction of the dose (77; 1.0%) and an increased dose (4; 0.1%). Overall, kanamycin (315; 13.5%), cycloserine (296; 12.7%), and pyrazinamide (240; 10.3%) were the most frequent medicines withdrawn due to ADRs. In terms of medicine–reaction pair, ear and labyrinth disorders were associated with kanamycin (185; 7.9%), and psychiatric disorders (181; 7.8%) and gastrointestinal disorders (93; 4.0%) were responsible for the withdrawal of cycloserine and ethionamide, respectively.

As indicated in the Appendix A, we also analyzed the most frequently withdrawn medicines for different age groups. Accordingly, psychiatric disorders associated with cycloserine (168; 8.8%) were more prominent in adults than in adolescents and the elderly (X^2^ = 1840.785, DF = 25, *p* < 0.0001).

## 3. Discussion

The aim of the present study was to evaluate the magnitude and characteristics of DR-TB-associated ADRs through an analysis of the ADRs reported in the global pharmacovigilance database (VigiBase). It was the first of its kind to be carried out at the global level using VigiBase^®^. Studies conducted previously on the safety of DR-TB focused either on a few medicines or were extracted from few countries’ PV databases, without considering the wider scope of DR-TB medicines used in different countries. In summary, the most relevant results found in this analysis were that half of the ADR reports of DR-TB medicines were attributed to pyrazinamide, ethionamide, cycloserine, bedaquiline, clofazimine, and linezolid. Among the most frequently reported ADRs, there were a few serious conditions, such as peripheral neuropathy or prolongation of QT (most cases were attributed to linezolid and bedaquiline). Furthermore, almost one-third of the reports described events that required withdrawal of the suspected medicine or a reduction in the dose; clearly, this affects adherence to the treatment and can indirectly trigger resistance. Finally, it should be highlighted that 40% of the analyzed reports of suspected events started more than two months after the onset of the treatment, which stresses the importance of remaining alert for the potential adverse effects of medicines, enabling their quick identification and thus avoiding as many treatment withdrawals as possible.

### 3.1. Medicines Associated with the Most Frequent ADRs

Of the 25 suspected individual medicine or fixed-dose combinations associated with ADRs, pyrazinamide, ethionamide, cycloserine, bedaquiline, clofazimine, and linezolid were responsible for more than 50% of the ADRs reported. This could be due to the new all-oral approach of WHO, the introduction of repurposed and new drugs, the removal of kanamycin and capreomycin [15,24,25]., and the reduced use of amikacin, ethionamide/prothionamide, and PAS [24]. In contrast, linezolid is used more often due to its benefits despite its frequent and serious adverse events [25].

### 3.2. Adverse Drug Reactions

In the present study, vomiting was the most frequently reported ADR, followed by arthralgia, nausea, neuropathy peripheral, and prolongation of electrocardiogram QT, a finding that is more or less similar to the findings from previous studies [20,26,27,28]. The same sets of ADRs were also reported in a four-year retrospective study conducted by Arif et al. [29]. Similarly, gastrointestinal disturbances (18.4%), psychiatric disorders (5.5%), arthralgia (4.7%), and hepatitis (3.9%) were the top five ADRs reported by Tae et al. [23]. Slight differences in the frequencies of ADRs could be due to different factors such as the sociodemographic background, which may contribute to the occurrence of ADRs [30].

Furthermore, we evaluated the timing of the ADRs’ occurrence, revealing that most ADRs appeared in less than three months. The most frequent ADRs, such as vomiting, arthralgia, and nausea, had a mean reaction onset of less than 3 months. With few exceptions, such as a new signal recently detected in Eritrea with the development of alopecia, which has a delayed time to onset [31], most of the drugs’ ADRs started to appear within the first 3 months of treatment [32]. Additionally, other studies also indicated that the average time from the initiation of the anti-TB treatment to developing drug-induced liver injury was estimated to be about 24 days [33,34]. This is similar to the results of a retrospective observational cohort study using pharmacovigilance data, which was conducted in Nigeria by Avong et al. [19]. However, a study conducted by Madan et al. on ADRs caused by second-line anti-tubercular drugs used in Nepal reported that the mean onset time of ADRs was 7.85 months [35]. Overall, it is possible to conclude that patients on DR-TB medicines mostly develop more frequent ADRs around the beginning of their treatment, and the ADRs will continue to appear throughout the duration of the treatment. Therefore, ADRs should be closely monitored for the entire duration of the treatment.

On the other hand, we also analyzed the seriousness of the ADR. Accordingly, almost half of the reports were serious ADRs mainly caused by bedaquiline, delamanid, clofazimine, linezolid, and cycloserine. Significant numbers of these ADRs were not recovered/not resolved when they were reported, while 12.1% were fatal. This is worrying, as an important proportion of the medicines responsible for serious ADRs are the backbone of the DR-TB treatment currently in use [13,14,15], and the outcomes of these ADRs are not pleasant.

### 3.3. Outcomes of the Reactions

The present study showed that almost one-quarter (23.3%) of the ADRs’ outcomes were fatal or not resolved when they were reported, while 1.1% had recovered/resolved with sequelae, and 25.4% of the ADRs were still recovering/resolving. Based on the WHO’s recommendations, it is very important to strictly monitor the unfavorable effects of medicines and put an effective PV system in place when administering new and repurposed DR-TB medicines [36,37]. As indicated in Figure 4, elderly patients have poorer outcomes to the reaction relative to adolescents and adults. The susceptibility of older people to ADRs is considered to be secondary both to extrinsic factors (prescription and management of medication) and intrinsic factors (pharmacokinetics and pharmacodynamics) [38]. Overall, vulnerability to ADRs is more probable at older ages—especially at a hepatotoxic level—due to a significant reduction in the clearance rate of metabolized drug agents by the cytochrome P450 enzyme, changes in the distribution of hepatic blood flow, and other factors affecting liver function [39].

Another key finding of this study showed that of the 7474 reports, 31.2% required complete withdrawal of the suspected medicine(s), followed by a reduction in the dose (1.03%) and an increase in the dose (0.05%). Overall, kanamycin (13.5%), cycloserine (12.7%), and pyrazinamide (10.3%) were the most frequent medicines withdrawn due to ADRs. In terms of medicine–reaction pairs, ear and labyrinth disorders were associated with kanamycin (7.9%), and psychiatric disorders (7.8%) and gastrointestinal disorders (4.0%) were responsible for the withdrawal of cycloserine and ethionamide, respectively. Similarly, studies conducted in South Africa, Nigeria, China, and India indicated that a significant proportion of patients required permanent discontinuation of the offending drug associated with ADRs [19,40,41,42] A retrospective study conducted in South Korea indicated that PAS was withdrawn in 11.3% of the patients due to uncontrolled diarrhea, nausea, or vomiting [23]. According to Bhatt et al., cycloserine produced major adverse psychotic reactions, which led to the discontinuation of the treatment [20]. To improve patients’ quality of life, tolerability, and thus treatment success, early diagnosis and appropriate management of ADRs are important. Serious ADRs may lead to poor adherence and discontinuation of the treatment before smear conversion, which might lead to the transmission of resistant strains of TB to the community [43].

### 3.4. Strengths and Limitations of the Study

The strength of this study is the use of VigiBase as a data source, which included reports submitted by multiple countries. This makes it a valuable resource for identifying safety concerns with medications that may not be apparent in any one country.

However, there are also some limitations. Firstly, VigiBase only includes data from reports submitted by the WHO’s PIDM countries. This means that there may be significant gaps in the data, particularly from countries that are not members of the WHO’s PIDM. Secondly, according to Haggar et al. [44], ICSRs from Africa make up <1% of the global total reports in VigiBase, and another study was conducted to analyze the global patterns of ADRs submitted to VigiBase over a decade, highlighting that low-income countries reported relatively more ADRs for anti-infective cases (including DR-TB cases) than high-income countries [45]. These findings imply that low-income countries known to report ADRs for anti-infective cases are less visible in VigiBase globally; therefore, this study should be supplemented with other data sources to address this gap. Thirdly, the data in VigiBase are not always complete. Large numbers of reports were excluded because of missing information such as the ADRs’ start date, the medicine’s start date, and unknown indications. This appears to be a challenge globally, as studies conducted in São Paulo (Brazil), western China, the Japanese AE database, Catalonia, the Midi-Pyrénées PV center (France), and Portugal on the completeness and quality of reporting showed similar trends [46,47,48]. We therefore strongly recommend that PIMD member countries and WHO-UMC should work towards improving the quality of the reports. Lastly, VigiBase does not include all of the possible ADRs collected by pharmacovigilance centers, as countries decide on what information can be shared with VigiBase. Similarly, since there are no data on the exposure denominator, and only individual case safety reports (ISCRs) of suspected adverse drug reactions (ADRs) that have already occurred are collected by the WHO Programme for International Drug Monitoring (PIDM)/UMC database, it is impossible to determine the risk associated with the suspected medicines. Given these limitations, it is important to consider additional sources of information when conducting research on the safety of medicines.

## 4. Methods and Materials

### 4.1. Data Source

This study was conducted on data obtained from VigiBase^®^, which is maintained by the WHO’s Uppsala monitoring center (WHO-UMC), Uppsala, Sweden. VigiBase is the WHO’s only global database of the reported potential side effects of medicinal products. It is the largest database of its kind in the world, with over 32 million reports of suspected ADRs submitted since 1968 by the member countries of the WHO’s Program for International Drug Monitoring (PIDM) [49]. It is a repository of individual case safety reports (ICSR) of ADRs collected by the national pharmacovigilance centers of about 152 member countries and 23 associate members [49].

VigiBase is linked to medical and drug classifications including terminologies such as the WHO-ART/MedDRA and WHO ICD classifications, which are vital to effective and accurate analyses [49,50].

Thus, this study included an analysis of all the suspected ADRs associated with DR-TB medicines included in VigiBase. Each report represents a single individual who may have experienced one or several ADRs simultaneously. As a result, the number of reported ADRs could be higher than the number of patients.

### 4.2. Variables and Definitions

The following ADR report variables were included in the data analysis: medicines, country of origin, age, patients’ age group, MedDRA preferred terms (PT) and system organ class (SOC), the seriousness of the ADR, the outcome of the reaction, and action taken. Reports with unknown information for a particular variable were included in the analysis to obtain a complete panorama of the data.

An adverse drug reaction (ADR) is a response to a medicinal product, which is noxious and unintended. Adverse reactions may arise from use of the product within or outside the terms of the marketing authorization or from occupational exposure. Uses outside the marketing authorization include off-label use, overdose, misuse, abuse, and medication errors [51].

An indication was defined as a DR-TB case that led to a treatment being recommended. The following indications/cases were included in the analysis: extensively resistant pulmonary tuberculosis, multidrug-resistant tuberculosis, rifampicin-resistant tuberculosis, multidrug-resistant pulmonary tuberculosis, mono-resistant tuberculosis, poly-resistant tuberculosis, mono-resistant pulmonary tuberculosis, poly-resistant pulmonary tuberculosis, and multidrug-resistant pulmonary tuberculosis.

Serious ADRs: According to the ICH E2A guidelines, a serious adverse event or reaction is an AE that either leads to death or a life-threatening experience, to hospitalization or prolongation of hospitalization, to persistent or significant disability, or to a congenital anomaly or that is a medically important event or reaction [52].

The patient’s age group was the patient’s age at the ADR’s onset. The age groups in this study were categorized into children, adolescents (0–18 years old), adults (19–64 years old), and the elderly (above 65 years old).

The MedDRA SOCs are the 27 groups of ADR terms at the top of the MedDRA hierarchy, which use groupings by etiology (e.g., infections and infestations), manifestation site (e.g., gastrointestinal disorders), or purpose (e.g., surgical and medical procedures) [50].

The MedDRA preferred terms (PTs) are discrete descriptors (single medical concepts) for a symptom, sign, disease diagnosis, therapeutic indication, investigation, surgical or medical procedure, or medical, social, or family history characteristic [50].

According to the MedDRA SOC, an “investigation” is a clinical laboratory test (including biopsies), radiologic test, physical examination parameter, or physiologic test (e.g., a pulmonary function test).

### 4.3. Search Criteria

The search criteria used in this investigation are summarized in Table 6 below.

For each report describing ADRs attributed to any of these medicines, detailed information regarding the patient’s demographics (age, sex, country, and medical history), drugs (an indication of use, route of administration, and start and end date), ADRs (date of onset, seriousness, outcome, outcomes of dechallenge and rechallenge, and causality), and administrative information (type and source of the report) was recorded. A graphic flowchart of ADRs selection is illustrated in Figure 6.

### 4.4. Data Analysis

A descriptive analysis was performed on the eligible reports from VigiBase using prespecified indicators. The ADRs were stratified by sex (male or female), age group (<18 years, 18–64 years, and ≥ 65 years), reporting country, seriousness (serious and non-serious), outcome of the reaction (fatal, not recovered/not resolved, recovered/resolved, recovered/resolved with sequelae, and recovering/resolving), and dechallenge and rechallenge. A combination of Excel and SPSS data analysis software [53] was utilized for data cleaning and data analysis.

## 5. Conclusions

In conclusion, 39 countries’ reports were included in this study. The majority of the reports were submitted by India, followed by South Africa and Eswatini. The results of this study confirmed that pyrazinamide, ethionamide, cycloserine, bedaquiline, clofazimine, and linezolid were responsible for more than half of the ADRs reported in VigiBase during the period from January 2018 to December 2020. Vomiting was the most frequently reported ADR, followed by arthralgia, nausea, peripheral neuropathy, and prolongation of electrocardiogram QT. Elderly patients (aged more than 65 years) had poorer outcomes of the reaction relative to adolescents (0–18 years) and adults (19–64 years). We would like to emphasize that strict monitoring of the unfavorable effects of medicines and effective pharmacovigilance systems should be put in place, as a high proportion of the medicines responsible for serious ADRs are the backbone of DR-TB treatment currently in use.

## Figures and Tables

**Figure 1 antibiotics-12-00811-f001:**
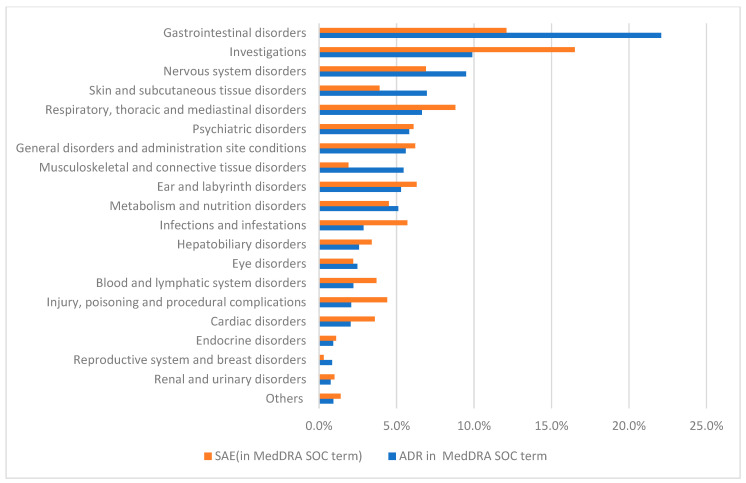
Frequency of the 7474 reported ADRs and the proportion of SAE according to the system/organ classification (SOC; MedDRA terms).

**Figure 2 antibiotics-12-00811-f002:**
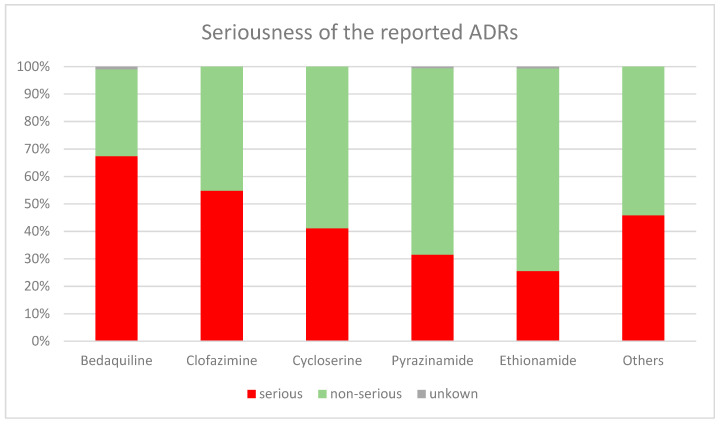
Graphical representation of top five reported medicines of causing serious ADRs.

**Figure 3 antibiotics-12-00811-f003:**
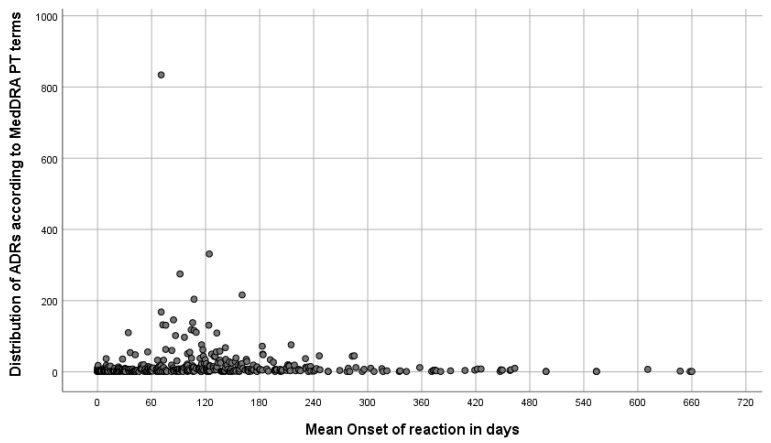
Mean onset of the reactions in days (dots represent ADRs).

**Figure 4 antibiotics-12-00811-f004:**
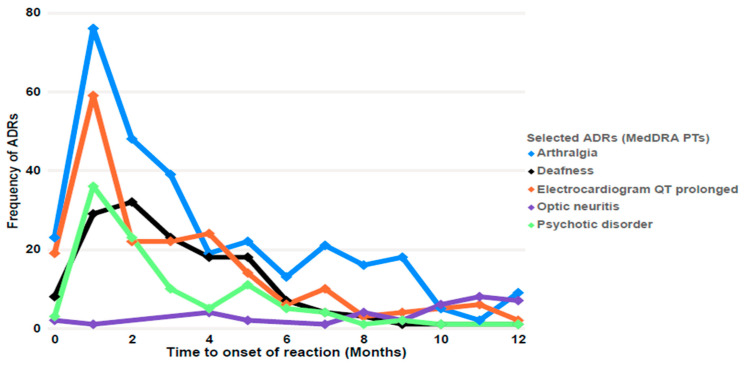
Time to the onset of the reaction of selected ADRs (based on MedDRA PT terms).

**Figure 5 antibiotics-12-00811-f005:**
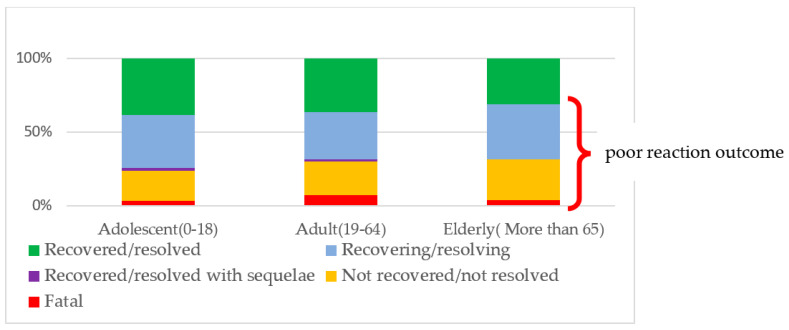
Outcomes of the reaction according to age group.

**Figure 6 antibiotics-12-00811-f006:**
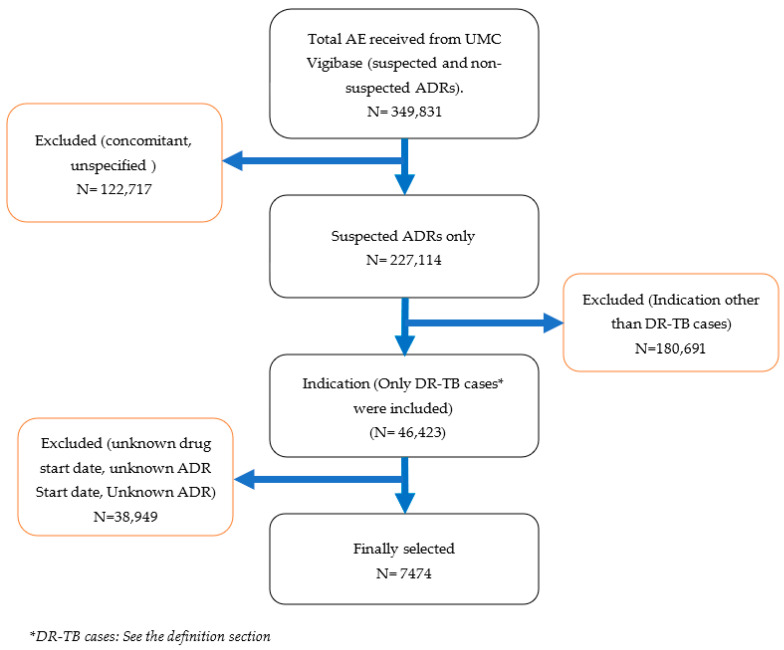
Schematic diagram of the selection criteria of ADRs from VigiBase data used to filter the records.

**Table 1 antibiotics-12-00811-t001:** Characteristics of the 7474 reports of adverse drug reactions (ADRs) attributed to medicines used in the management of DR-TB.

Indicators (*n* = 7474)		Frequency	%
Age group (years)	Children and adolescent (0–18)	652	8.7
	Adult (19–64)	6435	86.1
	Elderly (above 65)	387	5.2
Sex	Female	3444	46.1
	Male	4030	53.9
Seriousness of ADR (*n* = 7474)	Serious	3365	45.0
	Non-serious	4065	54.4
	Unknown	44	0.6
Route of administration	Oral	6013	80.5
	Intramuscular	568	7.6
	Intravenous	110	1.5
	Other	8	0.1
	Unknown	775	10.4
Dechallenge action	Dose not changed	3568	47.7
	Drug withdrawn	2334	31.2
	Dose reduced	77	1.0
	Dose increased	4	0.1
	Unknown	818	10.9
	Not applicable	673	9.0
Dechallenge outcome	Reaction abated	3854	51.6
	Effect unknown	2006	26.8
	No effect observed	1274	17.1
	Fatal *	340	4.6
Rechallenge action	Unknown	6703	89.7
	Rechallenge	771	10.3
Rechallenge outcome	Effect unknown	7331	98.1
	No recurrence	104	1.4
	Reaction recurred	39	0.5
Outcome of the reaction	Recovered/resolved	2113	28.3
	Recovering/resolving	1901	25.4
	Recovered/resolved with sequelae	82	1.1
	Not recovered/not resolved	1332	17.8
	Fatal **	408	5.5
	Unknown	1638	21.9

Fatal *, fatal reaction outcome after withdrawal of the treatment; Fatal **, overall fatal outcome of the reaction.

**Table 2 antibiotics-12-00811-t002:** Seriousness of the 7474 reported ADRs according to the suspected medicines, age group, and specific ADRs.

Variables	Medicine	Seriousness					
		Yes (*n* = 3365)		No (4065)		Unknown (44)	
		N	%	N	%	N	%
Age group (years) *	Children and adolescent (0–18)	363	55.7%	287	44.0%	2	0.3%
	Adult (19–64)	2822	43.8%	3571	55.5%	42	0.6%
	Elderly (above 65)	180	46.5%	207	53.5%		0.0%
Top five ADRs according to the SOC MedDRA terms	Respiratory, thoracic, and mediastinal disorders (*n* = 496)	296	59.7%	200	40.3%	-	-
	Psychiatric disorders (*n* = 435)	206	47.4%	223	52.2%	6	1.4%
	Nervous system disorders (*n* = 709)	231	32.6%	471	66.4%	7	0.01%
	Skin and subcutaneous tissue disorders (*n* = 520)	131	25.2%	387	74.4%	2	0.01%
	Gastrointestinal disorders (*n* = 1650)	406	24.6%	1233	74.7%	11	0.6%
	Others (*n* = 3646)	2095	57.1%	1551	42.3%	18	0.5%

* chi-square = 34.9456; *p* ≤ 0.00001.

**Table 3 antibiotics-12-00811-t003:** Top five reported medicines and the top five reported serious ADRs based on MedDRA SOCs for these medicines.

	Gastrointestinal Disorders	Respiratory, Thoracic, and Mediastinal Disorders	Nervous System Disorders	Ear and Labyrinth Disorders	General Disorders and Administration Site Conditions	Others	Total N (%)
Linezolid, *n* (%)	34 (9.9)	33 (9.6)	43(2.3)	-	22(6.4)	210 (61.4)	342 (100)
Delamanid, *n* (%)	33 (8.4)	44 (11.3)	23 (1.2)	3 (0.8)	34 (8.7)	254 (64.9)	391 (100)
Cycloserine, *n* (%)	26 (9.1)	18 (6.3)	25 (1.4)	4 (1.4)	14 (4.9)	199 (69.6)	286 (100)
Clofazimine, *n* (%)	48 (13.0)	50(13.5)	24 (1.3)	1 (0.3)	26 (7.0)	221 (59.7)	370 (100)
Bedaquiline, *n* (%)	40 (8.8)	59 (12.9)	25 (1.4)	1 (0.2)	32 (7.0)	300 (65.6)	457 (100)
Others, 1519 (%)	225 (14.8)	92 (6.0)	91 (5.9)	203 (13.3)	81 (5.3)	827 (54.4)	1519 (100)
Total, 3365 (%)	406 (12.1)	296(8.8)	231 (6.8)	212 (6.3)	209 (6.2)	2011 (59.7)	3365 (100)

**Table 4 antibiotics-12-00811-t004:** Classifications of the reported serious ADRs according to the MedDRA SOC and PT terms.

Top Five Reported Serious ADRs Based on MedDRA SOC Terms (*n* = 3365)	The Most Frequent Individual ADRs Based on MedDRA PT Terms
Ear and labyrinth disorders (*n* = 212)	Deafness (100; 47.2%), tinnitus (31; 14.6%), hypoacusis (26; 12.3%), and others (55; 25.9%)
Gastrointestinal disorders (*n* = 406)	Vomiting (222; 54.6%), gastritis (63; 15.5%), nausea (40; 9.8%), and others (81; 19.9%)
Nervous system disorders (*n* = 140)	Peripheral neuropathy (70; 50%), headache (32; 22.8%), optic neuritis (28; 20%), and others (10; 7.1%)
Respiratory, thoracic, and mediastinal disorders (*n* = 296)	Dyspnea (93; 31.4%), respiratory failure (37; 12.5%), cough (30; 10.1%), and others (136; 45.9%)
General disorders and administration site conditions (*n* = 209)	Pyrexia (41; 19.6%), chest pain (37; 17.7%), asthenia (22; 10.5%), and others (100; 47.8%)
Others (*n* = 2011)	Electrocardiogram QT prolonged (135; 6.7%), anemia (91; 4.5%), psychotic disorders (73; 0.6%), and others (1712; 85.5%)

**Table 5 antibiotics-12-00811-t005:** Cross-tabulation of ADR outcomes against seriousness of reaction.

		Outcome of the Reaction						Total
		Fatal	Not Recovered/Not Resolved	Recovered/Resolved	Recovered/Resolved with Sequelae	Recovering/Resolving	Unknown	
Seriousness	Yes	404 (12.0%)	435 (12.9%)	705 (20.9%)	44 (1.3%)	641 (19.0%)	1136 (33.7%)	3365 (100%)
	No	4 (0.1%)	884 (21.7%)	1384 (34.0%)	38 (0.9%)	1260 (30.9%)	495 (12.2%)	4065 (100%)
	Unknown	-	13 (29.5%)	24 (54.5%)	-	-	7 (15.9%)	44 (100%)
Total		408 (5.45)	1332 (17.82%)	2113 (28.2%)	82 (1.1%)	1901 (25.4%)	1638 (21.9%)	7474 (100)

**Table 6 antibiotics-12-00811-t006:** Summary of search criteria used.

Search Criteria	
Substance	Amikacin, PAS, bedaquiline, capreomycin, cilastatin-imipenem combination, clofazimine, cycloserine, delamanid, ethambutol, ethionamide, kanamycin, levofloxacin, linezolid, meropenem, moxifloxacin, protionamide, pyrazinamide, streptomycin, terizidone, isoniazid, rifampicin, amoxicillin–clavulanic acid combination, gatifloxacin, thioacetazone, ethambutol–isoniazid–pyrazinamide–rifampicin (rhze), isoniazid–pyrazinamide–rifampicin (rhz), isoniazid–rifampicin (rh), ethambutol–isoniazid (eh), and isoniazid–thioacetazone.
Drug involvement	Suspected
Terminology	MedDRA
Reactions	All
Country	All
Years	1 Jan 2018 to 31 December 2020
Other	Search in indication and medical history (coded fields), Search terms (MedDRA): “extensively resistant pulmonary tuberculosis”, “multidrug resistant tuberculosis”, “rifampicin resistant tuberculosis”, “multidrug resistant pulmonary tuberculosis”, “mono-resistant tuberculosis”, “poly-resistant tuberculosis”, “mono-resistant pulmonary tuberculosis”, “poly-resistant pulmonary tuberculosis”, “multidrug resistant pulmonary tuberculosis”, “other respiratory tuberculosis, other”. Reports are selected: if any of the terms above appears in indication or medical history; additionally, reports where indication is missing will be included.
De-duplicated dataset?	Yes

## Data Availability

The datasets used during this study are not publicly available due to agreements between the data contributors to the database used (VigiBase) and the database’s custodian. The national centers of the WHO’s Program for International Drug Monitoring (PIDM) contribute data to VigiBase, and the Uppsala Monitoring Centre serves as the custodian in its capacity as a WHO collaboration center for international drug monitoring. On reasonable request, the corresponding author may make some subsets of the data available.

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
