# Peer review of "Safety Profile of Medicines Used for the Treatment of Drug-Resistant Tuberculosis: A Descriptive Study Based on the WHO Database (VigiBase®)"

_antibiotics, 2023, doi:10.3390/antibiotics12050811_

Round 1
Reviewer 1 Report
The paper aimed to describe the magnitude and characteristics of DR-TB 16 related ADRs through analysis of ADRs reported to the WHO database in the period of January 2018 to December 2020; and found that A high proportion of the medicines responsible for frequent ADRs are the backbone of the shorter DR-TB treatment regimen currently in use. The authors understand the limitation of the study. The paper could provide interesting information to audience. The manuscript needs more proofreading. The following comments aim to improve the manuscript.
Minor:
1. In Table 1, how did the authors determine the Seriousness of ADR, please describe it at the table legend?
1. Please correct “fol-lowed by ethionamide” (line 23)
2. Please correct “fre-quent ADRs are the” (line 27)
3. “medicines used in patients with DR TB is” acronym is not consistent some are written as “DR-TB” (line 72)
4. Please modify “Out the 25 different medicines” (line 90)
Reviewer 2 Report
Research Manuscript Title: Safety profile of medicines used for the treatment of Drug resistance Tuberculosis: A descriptive study based on the WHO database(VigiBase®)
Manuscript ID: Antibiotics-2306343
The purpose of this study was aimed at describing the magnitude and characteristics of DR-TB related ADRs through analysis of ADRs reported to the WHO database (VigiBase® ) in the period of January 2018 to December 2020.
Major comments
There are some issues which need to be addressed by the authors:
Comment #1: In this study, how many patients received bedaquiline and delamanid concomitantly ? Were there any fatal drug-related adverse events arising out of sudden cardiac death(SCD) that occurred among patients ? How many patients were pregnant while receiving antitubercular treatment & what were the outcomes ?
Comment #2: The “Data Search Criteria” can preferably be arranged in a tabular form.
Comment #3: English language needs major touching up. Many sentences are confusing, do not lead to scientific meaning.
Round 2
Reviewer 2 Report
All comments have been addressed.